# Investigation of the New Inhibitors by Sulfadiazine and Modified Derivatives of α-D-glucopyranoside for White Spot Syndrome Virus Disease of Shrimp by In Silico: Quantum Calculations, Molecular Docking, ADMET and Molecular Dynamics Study

**DOI:** 10.3390/molecules27123694

**Published:** 2022-06-08

**Authors:** Ajoy Kumer, Unesco Chakma, Md Masud Rana, Akhel Chandro, Shopnil Akash, Mona M. Elseehy, Sarah Albogami, Ahmed M. El-Shehawi

**Affiliations:** 1Laboratory of Computational Research for Drug Design and Material Science, Department of Chemistry, European University of Bangladesh, Dhaka 1216, Bangladesh; 2Department of Chemistry, Bangladesh University of Engineering and Technology, Dhaka 1000, Bangladesh; 3Department of Electrical and Electronics Engineering, European University of Bangladesh, Gabtoli, Dhaka 1216, Bangladesh; unesco@eub.edu.bd; 4Department of Fishing and Post Harvest Technology, Sher-e-Bangla Agricultural University, Dhaka 1207, Bangladesh; rana.fpht@sau.edu.bd; 5Department of Poultry Science, Faculty of Animal Science & Veterinary Medicine, Sher-e-Bangla Agricultural University, Dhaka 1207, Bangladesh; akhel.sau6012@gmail.com; 6Department of Pharmacy, Daffodil International University, Sukrabad, Dhaka 1207, Bangladesh; shopnil29-059@diu.edu.bd; 7Department of Genetics, Faculty of Agriculture, University of Alexandria, Alexandria 21545, Egypt; monaahmedma@yahoo.com; 8Department of Biotechnology, College of Science, Taif University, P.O. Box 11099, Taif 21944, Saudi Arabia; dr.sarah@tu.edu.sa (S.A.); elshehawi@hotmail.com (A.M.E.-S.)

**Keywords:** DFT, HOMO, LUMO, docking, molecular dynamic, WSSV, ADMET

## Abstract

The α-D-glucopyranoside and its derivatives were as the cardinal investigation for developing an effective medication to treat the highest deadly white spot syndrome virus (WSSV) diseases in Shrimp. In our forthcoming work, both computational tools, such as molecular docking, quantum calculations, pharmaceutical kinetics, ADMET, and their molecular dynamics, as well as the experimental trial against WSSV, were executed to develop novel inhibitors. In the beginning, molecular docking was carried out to determine inhibitors of the four targeted proteins of WSSV (PDB ID: 2ED6, 2GJ2, 2GJI, and 2EDM), and to determine the binding energies and interactions of ligands and proteins after docking. The range of binding affinity was found to be between −5.40 and −7.00 kcal/mol for the protein 2DEM, from −5.10 to 6.90 kcal/mol for the protein 2GJ2, from −4.70 to −6.2 kcal/mol against 2GJI, and from −5.5 kcal/mol to −6.6 kcal/mol for the evolved protein 2ED6 whereas the L01 and L03 display the highest binding energy in the protein 2EDM. After that, the top-ranked compounds (L01, L02, L03, L04, and L05), based on their high binding energies, were tested for molecular dynamics (MD) simulations of 100 ns to verify the docking validation and stability of the docked complex by calculating the root mean square deviation (RMSD) and root mean square fluctuation (RMSF). The molecules with the highest binding energy were then picked and compared to the standard drugs that were been applied to fish experimentally to evaluate the treatment at various doses. Consequently, approximately 40–45% cure rate was obtained by applying the dose of oxytetracycline (OTC) 50% with vitamin C with the 10.0 g/kg feed for 10 days. These drugs (L09 to L12) have also been executed for molecular docking to compare with α-D-glucopyranoside and its derivatives (L01 to L08). Next, the evaluation of pharmacokinetic parameters, such as drug-likeness and Lipinski’s principles; absorption; distribution; metabolism; excretion; and toxicity (ADMET) factors, were employed gradually to further evaluate their suitability as inhibitors. It was discovered that all ligands (L01 to L12) were devoid of hepatotoxicity, and the AMES toxicity excluded L05. Additionally, all of the compounds convey a significant aqueous solubility and cannot permeate the blood-brain barrier. Moreover, quantum calculations based on density functional theory (DFT) provide the most solid evidence and testimony regarding their chemical stability, chemical reactivity, biological relevance, reactive nature and specific part of reactivity. The computational and virtual screenings for in silico study reveals that these chosen compounds (L01 to L08) have conducted the inhibitory effect to convey as a possible medication against the WSSV than existing drugs (L09, L10, L11 and L12) in the market. Next the drugs (L09, L10, L11 and L12) have been used in trials.

## 1. Introduction

The fisheries sector has to play an essential role in filling the demand for proteins from the large population of not only Bangladesh [1], but also the whole world with ensuring food security [2], poverty alleviation [3,4], and the national economy [5,6]. In the contemporary era, about 60% of animal proteins for human beings come from river fish or sea fish [7], As well, as 3.50% of our Gross Domestic Product (GDP), (Bangladesh) [8]. In addition, the 25.71% of GDP in the agricultural sectors has been covered by the fisheries sector [9]. Now, Bangladesh has attained self-sufficiency in the production of fishes with respect to the various species, specially cultivated fishes, as well as developed storage systems to exceed its demands [10]. Millions of people in our country, especially rural people, are directly or indirectly involved in fish production [4,11], harvesting, transportation, marketing, and processing, which have opened new employment doors for more than 8.0 million people per year in Bangladesh [12]. The fisheries sector of Bangladesh brought unparalleled success globally, placing fifth in the world of fish production [13,14]. However, it may be revealed that Bangladesh is not only lagging the exporting fish and fish products, but also earning foreign currency. According to the 2018–2019 financial report, USD 455 million was earned by exporting 68,655.00 tons of fish and fish products, and 43.38% (31158.0 tons) of this contribution came from shrimp [15]. That is why shrimp (*Penaeus monodon*), called the "white gold" of Bangladesh in recent decades [16], has turned the wheel for fortune of the people of the southern part of our country [17]. Bangladesh has more than 58,000.0 marine shrimp farms; the average size ranges between 3.50 and 4.00 hectors for production, although single ponds as large as 45.0 hectors still exist. At present, the total area of shrimp and prawn farms is about 258,553.0 hectares. Most farms are located in the Khulna, Satkhira, Bagerhat, Cox’s Bazaar, Chittagong, Barguna, and Bhola districts [18]. Shrimp production in Bangladesh has been increasing day by day since the last couple of decades. Prior to 2018–2019, the total production of shrimp/prawns was about 239855.0 metric tons, and 40,000.00 metric tons of shrimp have been exported to different countries around the world. According to the Export Promotion Bureau (EPB), Bangladesh earned USD 4.36 million by exporting shrimp during the financial year 2019–20 [19].

Shrimp farming is a very lucrative and gratifying business for earning money, or even foreign remittance in Bangladesh and worldwide, and it is considered the second largest foreign exchange earning asset in our country [20]. However, this farming is going the downward trend day by day due to poor management, environmental impact, cost impact, over-salinity, unexpected flooding, and lack of scientific functions last few years [17,21]. Additionally, certain pathogenic diseases caused by viruses, bacteria, and fungi were found recently, which is considered the towering and cardinal threat for Shrimp farming [22]. Among them, the white spot disease, detected in 2001 in Bangladesh [23], was considered a warning to the shrimp farmers of Bangladesh. It may cause the full destruction of Shrimp farming for accelerating the death of the host body [24,25]. White spot disease is a viral disease of penaeid shrimp and is employed by the white spot syndrome virus (WSSV) [26,27], which is highly lethal and contagious, causingdeath in a short time. Moreover, WSSV is a DNA virus recognized as the only member of the genus Whispovirus (family Nimaviridae) [28]. Generally, the WSSV infects numerous cells from shrimps’ ectodermal and mesodermal origin; pathogenesis involves widespread tissue necrosis and disintegration, which can be diagnosed by quantitative analysis from a polymer chain reaction (PCR) [29]. WSSV is the most deadly of the more than 20 viruses that can affect penaeid shrimp worldwide. The first outbreak of the white spot disease was reported in a shrimp farm in Taiwan in 1992, followed by other shrimp farms in China, Thailand, Korea, Malaysia, and east and southeast Asia [30]. In India, WSSV was first detected in tiger shrimp farms across a wide range from Andhra Pradesh to Sirkali in Tamilnadu in 1994 [31,32]. In Bangladesh, white spot disease was found in 2007 and gradually increasing day by day [22]. The disease can find on the Shrimp’s body at the age of 30–60 days after the release of shrimp fry. During the first stage, there are no external symptoms of the disease, but, 3 or 4 days later, the severity of the illness increases, obtaining the symptoms, such as white spots on the skin and a reddened head. It acts as the uncanny and devastating disease with 100% mortality within 3–10 days of the appearance of clinical signs in tiger shrimp. Around the world, more than USD 1.0 billion has been lost annually due to WSSV. With no treatment or prevention method with clinical proof or an approved medicine, the white gold will disappear in the near future, not only from Bangladesh but also from other countries. Now, it is the demand of time to search a new remedy or treatment for WSSV; otherwise, Bangladesh will lose 5.0% of its GDP from shrimp farming with the producing of unemployment vacancy about 8.0 million people in coastal areas.

No perfect cure or treatment drugs have ever been found for the WSSV disease of shrimp, and poor scientific research and progress are also responsible for spreading this disease. However, this research project has designed to search for a new potential drug to treat the deadly WSSV disease through computational tools, such as molecular docking; their molecular dynamic; drug-protein complex interactions; action mechanisms; and an ADMET study, which have already established that the most acceptable method for designing for new inhibitor by an in silico study. Sulfadiazine and its derivatives have already used as an antibiotic for numerous viral disease treatments for fish and fisheries farms [33,34]. Subsequently, the derivatives of α-D-glucopyranoside have been reported in recent years acting as antimicrobial potential drugs, antifungal, antibacterial, and antiviral drugs, as well as having an antifungal potential for white and black fungus. As the WSSV disease of shrimp is a viral infection; sulfadiazine and its derivatives have been selected, as well as derivatives of α-D-glucopyranoside, for an in silico study to convey their antiviral activities. Firstly, the PASS prediction was performed to isolate initial profiles of compounds. Next molecular docking was done to evaluate their binding capacities. The molecular docking was validated by the molecular dynamic. Finally, this study conveys the quantum properties and ADMET parameters.

## 2. Computational Details of Procedure

### 2.1. Optimization and Ligand Preparation

Material Studio 8.0 was utilized for computational models and hypothetical investigation, whereas the DFT functional of the DMol3 code with DNP basis set was applied for determination of the chemical descriptor, quantum properties and geometrical optimization [35,36,37]. Making use of the DFT functional, the quantum properties, such as εLUMO, εHOMO, energy gap (ΔE gap), ionization potential (I), electron affinity (A), chemical potential (µ), electro negativity (χ), hardness (η), softness (s) and electrophilicity (ω) were calculated by following Equations (1)–(8): (1)$${\;E}_{\mathrm{gap}}=(E_{\mathrm{LUMO}}-E_{\mathrm{HOMO}})$$
(2)$$I=-E_{\mathrm{HOMO}}$$
(3)$$A=-E_{\mathrm{LUMO}}$$
(4)$$\left(\chi \right)=\frac{I+A}{2}$$
(5)$$\left(\omega \right)=\frac{\mu^{2}}{2\eta }$$
(6)$$\left(µ\right)=-\frac{I+A}{2}$$
(7)$$\left(\eta \right)=\frac{I-A}{2}$$
(8)$$\left(S\right)=\frac{1}{\eta }$$

### 2.2. Protein Preparation and Collection

The four proteins (2ED6, 2GJ2, 2GJI, and 2EDM) with several entities of protease structures for WSSV were taken from the protein data bank (PDB), (http://www.rcsb.org, accessed on 21 February 2022). The 2ED6 is the envelope protein for WSSV, and other three proteins are the main proteases of WSSV. All of these proteins were found in shrimp evaluated through the X-ray diffraction method with high stable configuration Ramachandran outliers listed Table 1. After taking the proteases from the PDB, these were optimized in Discovery Studio via minimum energy. Water and heteroatoms were removed, and saved in PDB form for further works.

### 2.3. Molecular Docking and Visualization of Docking

PyRx software was employed to execute molecular docking, with the auto docking, after acquiring the necessary parameters (listed in Table 2) to assess the binding affinity of the ligand and protein to individual macromolecules. In the case of molecular docking, the protein was loaded as a macromolecule, and the ligand was also loaded as a ligand. After loading the ligand, this ligand was optimized with maximum energy and grid surface area which consists of centre of grid and dimension of grid. Next, it is maintained that the total surface area of ligand and protein were covered. The docking employment was executed with the parameters of Table 2 from the PyRx software of auto dock option. After the docking procedure, the Discovery Studio visualization was assessed for non-covalent interactions of the ligand-protein docking complex [40]. 

### 2.4. Pharmacokinetics and ADMET Studies

The absorption, distribution, metabolism, excretion and toxicity is expressed in shortage by ADMET, which are valuable and necessary factors in the drug development process [41,42]. The ADMET criterion was obtained by use of the SwissADME and pkCSM online tool: http://biosig.unimelb.edu.au/pkcsm/prediction_single/adme_1643650057.59 (accessed from the 10 January 2022) [43]. The AMES toxicity, blood-brain barrier (BBB), water solubility, total clearance, etc. are the primary key for development of a theoretical comparison of derivatives.

### 2.5. Lipinski Rule and Pharmacokinetics

SwissADME was used to forecast the pharmacokinetics, drug-likeness metrics; http://www.swissadme.ch/index.php [44], which is an online database accessed in 09 November 2021, is highly influential as well as adaptable properties of gaining access to information. The pharmacokinetics, including the topological polar surface area (TPSA) Å^2^, molecular weight, hydrogen bond donors (HBD), hydrogen bond acceptors (HBA), bond rotation numbers (NRB), lipophilicity were calculated for explain the drug likeness properties of ligands.

### 2.6. Molecular Dynamic

On a desktop or a high configuration laptop computer, molecular dynamic (MD) simulations were employed with the help of the NAMD application, which can be conducted interactively via live view or in batch mode [45]. The MD simulation was devoted to underpinning the docking results gained for the best drug proteins up to 100 ns for the holo-form (drug-protein) applying AMBER14 force field [46]. The whole process was equilibrated using a concentration of 0.9 percent NaCl at 298 K temperature. During the simulation, a cubic cell was propagated within 20 Å on each side of the process and under periodic boundary conditions. After the simulation, the RMSD and RMSF were analyzed using the VMD software.

### 2.7. White Spot Trial Procedure

In the case of WSSV testing in the pond, 10 g of tablets was mixed with the shrimp feed as daily meals. In this trial, 10 g of tablets in 1.0 kg of shrimp feed was mixed for distribution in the ponds for seven days. Next, 10 g of vitamin C per kg of feed with 10 g of drugs was with feed was put in ponds as daily meal for ten days.

## 3. Results and Discussions

### 3.1. Optimized Structure

Optimized molecular structure is an important structural geometry for the study of a computational procedure to determine the quantum calculations of any chemical species [47]. In addition, the most stable configuration of any chemical structure possesses the accurate calculation of computational parameters. All compounds in this study were computationally optimized using the DFT functional, and their primary and most stable configuration with the low energy required for optimization is observed. The Methyl-4,6-O-benzylidene-2-O-(2,6-dichlorobenzoyl)- α-D-glucopyranoside (L01), Methyl-4,6-O-benzylidene-2-O-(2,6-dichlorobenzoyl)-3-O-pivaloyl-α-D-glucopyranoside(L02),Methyl-4,6-O-benzylidene-2-O-(2,6-dichlorobenzoyl)-3-O-(4-t-butylbenzoyl)-α-D-glucopyranoside(L03),Methyl-4,6-O-benzylidene-2-O-(2,6-dichlorobenzoyl)-α-D-glucopyranoside(L04),Methyl-4,6-O-benzylidene-2-O-(2,6-dichlorobezoyl)-3-O-pentanoyl-α-D-glucopyranoside(L05),Methyl-4,6-O-benzylidene-2-O-(2,6-dichlorobenzoyl)-3-O-hexanoyl-α-D-glucopyranoside(L06),Methyl-4,6-O-benzylidene-2-O-(2,6-dichlorobenzoyl)-3-O-lauroyl-α-D-glucopyranoside(L07),4,6-O-benzylidene-2-O-(2,6-dichlorobenzoyl)-3-O-myristoyl-α-D-glucopyranoside (L08), Sulfadiazine-d4 (L09), p-Mercapto-sulfadiazine (L10), Oxytetracycline (L11), and Oxytetracycline Hydrate (L12) are shown below in Figure 1.

### 3.2. HOMO, LUMO, and Chemical Reactivity Descriptors

Calculations of εLUMO, εHOMO, ΔE gap, chemical potential (µ), electronegativity (χ), hardness (η), softness (s), and electrophilicity (ω) for the compounds are listed in Table 3. The DFT functional was used to calculate these statistical profiles. The HOMO-LUMO energy gap is used to measure the chemical susceptibility in molecules. A wider HOMO-LUMO energy gap signifies that a molecule is highly unreactive and chemically unstable. The main reason for this is that the electronic transition is hindered due to a large energy gap from ground state to excited state. In general, a narrow HOMO-LUMO gap suggests that a molecule is highly stable [48,49,50,51,52,53,54]. According to the results in Table 3, the HOMO–LUMO gaps range between 6.630 eV and 7.990 eV for all of the studied chemicals, whereas the L04, L08, and L09 have minor energy gaps and minimum softness values. In contrast, the ligand L09 has the greatest hardness significance and the largest energy gap. It is observed that the order of the energy gap is L09>L08>L04>L05>L07>L11>L02>L06>L01>L03>L12>L10. Table 3 illustrates that the softness value is approximately 0.228 or less than 0.30. It is important to note that if an element’s softness level is more remarkable of small value, it will take less time to disintegrate and will degrade at a faster rate than others. Conversely, hardness is an essential attribute of a material whose measurements reflect its stability [55,56,57]. In general, the higher the hardness value of these compounds, the more strongly the molecules resist changes in electron configuration.

### 3.3. Frontier Molecular Orbital: HOMO and LUMO

The frontier molecular orbital (FMO) was used to assess the kinetics, and the engaged regions where the protein could be folded become the active pharmacophore or active functional group. The dark lemon color indicates the positive terminal of the orbitals in LUMO, while the pink color denotes the negative node. The more minor energy gap assists in the development of drug interaction with a protein. Alternatively, the yellow color for HOMO indicates the positive node of the orbital, and the light greenish color expresses the negative node of the orbital. From the picture in Figure 2, the most important fact obtained about the compounds is that the LUMO is located in the position of the chlorine-containing benzene ring of these derivatives. Conversely, the HOMO is located in the benzene ring in the end side and has no functional atoms or groups. That is why it must be said that the chemical and biological activities are controlled by the benzene ring and its functionalized atom as the side chain in this benzene ring.

### 3.4. Molecular Docking

To authenticate the pharmacological findings obtained, molecular docking simulations were carried out, and attest to the ligands’ binding of therapeutic compounds with the associated peptide vs. The four most available white spot disease (WSD) proteins [58,59]. The hydrogen and hydrophobic bonds are the primary cause of binding with protein for WSSV and show the binding affinity by molecular docking, whereas protein-ligand interaction is crucial in structurally oriented drug development. There is a general consensus that docking scores of more than 6.00 kcal/mol indicate a standard drug [60,61].

In addition, the molecular docking is a tried-and-true approach for understanding how two molecules engage and identifying the appropriate ligand configuration to create a minimum energetic complex. Using in silico experiments, it was discovered that each of the drug compounds in Table 4 can show the excellent binding affinity to the target proteins for WSSV, with values in the range from −6.20 to −6.90 kcal/mol; whereas the highest docking scores of the L03, the L04 for an envelope protein, and the L03 for a main protein might be regarded as standard drugs, whereas these drugs(L01, L03, and L04) can show much higher binding affinities than standardly used drugs (Sulfadiazine-d4, p-Mercapto-sulfadiazine, Oxytetracycline, and Oxytetracycline Hydrate).

In Table 5, the inhibition constant and ligand efficiency are presented, which may be extracted from the Auto Dock Vina module from MG-tools’ software packet. In the case of the inhibition constant, its the value is always below 80.0 micromolar (µM), which ranks as the most acceptable molecule as drug. All compounds show a magnitude below 40.0 µM, which is a very small amount to activate their activity.

### 3.5. Protein-Ligand Interaction

For the purpose of developing a novel medicine, the most vital component to consider is the ligand-protein interaction through the forming of weak bonds or a covalent bond, which offers approximate information regarding the binding affinity or energy of substances with the proteins of micro pathogens. In order to better understand the relationship between the molecule and the protein associated with white spot disease, the bond distance was measured. According to substitute data, there are different sorts of bonds: H-bonds, halogen bonds, hydrophilic bonds, Van dar Waal bonds and hydrophobic bonds. Additionally, the sites bound by the ligand are identified in the protein. According to the results, the ligand L02 has the most binding sides, with an H-bond count of eight and six hydrophobic bonds against the PDB ID: 2GJ2 protein. The ligand, L03, has the second-highest number of binding sides, with an H-bond count of of four and a count of hydrophobic bonds of six. Alternatively, the second target protein reported the highest number of binding sides in the ligand L08 with an H-bond count of seven and a hydrophobic bond count of eight. Moreover, both 2GJI and 2EDM have the H-bonds and hydrophobic bonds to the main protease of WSSV.

### 3.6. Pharmacokinetics and Drug-Likeness Study

Pharmacological drug-likeness is a groundbreaking evaluation of the potential of a particular chemical used as an oral medicine with respect to bioavailability. It is estimated that nine out of twelve targeted medicines are not transparently changed due to their negative effect, resulting in significant medication costs, time, and human resources being wasted [62]. This problem occurs due to failure to identify the actual drug characteristics. However, by employing a new approach, Lipinski’s five-rule, it is possible to readily test the aspects of lead compounds, such as their bioavailability and G.I. absorption, among other things [63]. In this section, our reported compounds had superior bioavailability, as well as a high G.I. absorption score. However, the ligands L03, L07 and L08 had a lower G.I. absorptions and lower bioavailabilities. Finally, Lipinski’s five-rule was satisfied for L01, L04, L-05 and L06, but L02, L03, L07, and L08 were not satisfied due to large molecular weights. Therefore, it is suggested that the ligands are safe to use. Table 6 shows the final outcome of pharmacokinetics and drug likeness.

### 3.7. Pharmacokinetics and ADMET Studies

In silico pharmacokinetics and ADMET techniques are utilized to quantify physicochemical properties in the early phases of the drug development process to decrease costs, time, resources, and effort. The ADMET study on ligand (L01–L12) was performed with the aid of In silico approaches by SwissADME: http://www.swissadme.ch/index.php (database accessed in 9 November 2021) and pkCSM online tool: http://biosig.unimelb.edu.au/pkcsm/prediction_single/adme_1643650057.59 (accessed from the 10 January 2022) [43] which projected the components’ absorptions, distributions, metabolisms, excretions, and toxicities. Table 7 and Table 8 displays the results of the ADMET data analysis. According to the results, these particular molecules (L01–L12) may not cross the blood-brain barrier. The range of total clearance rate (renal and non renal) of all compounds was within 0.424–0.873 and the highest clearance rate was reported for the ligand L08. Furthermore, it has been reported that the ligand L03 and L06 may metabolized in the CYP2C9 inhibitor at the same time, the ligand L05 and L06 were metabolized in the CYP1A2 inhibitor. Alternatively, the Caco-2 permeability test is based on a well-established technique that evaluates the rate of flux of a substance through polarized Caco-2 cell monolayers, and wherein the evidence gathered may be helpful to forecast in vivo absorption of medications in a different circumstances [64]. In this point, the range of Caco-2 Permeability has obtained from 0.747 to 1.878, and it is clear that the ligand L03 has high Caco-2 Permeability capability.

### 3.8. Aquatic and Non-Aquatic Toxicity

Table 8 demonstrates the aquatic and non-aquatic toxicities, which are also critical in determining whether pharmaceuticals or materials are acceptable in the environment before and after usages. Syntheses of molecules were identified with high water solubility, showing a significant affinity for the aqueous phase. Chemical compounds (L06 and L08) have the highest tendency to dissolve in water (LogS = −4.681) and compound L04 has the lowest dissolving tendency of all of the compounds studied so far (Log S = −5.509 and −5.021). All of the chemicals are devoid of hepatotoxicity, which implies that they will not create liver toxicity in humans or experimental animals when used as directed. The majority of medicines are free from AMES toxicity excluding compound L05. Moreover, the oral rat chronic toxicity range was reported within 1.497–10.30, where the highest chronic toxic compound was obtained for the ligand L05. Conversely, lethal doses ranging from 2.234 to 3.264 mol/kg in non-aquatic animals, such as oral rat acute toxicity (LD50), were found in Ligand L04 (3.264).

### 3.9. Protein-Ligand Interaction

The ligand binding site with receptor was identified with the help of Discovery Studio version 2020, and graphically represent on Figure 3, Figure 4 and Figure 5. In this case, at first, auto docking has been performed on the protein and ligand to identify the binding sites and obstructing the active site, as well as determining the amino acid residue [65,66]. The mostly present bond is hydrogen and hydrophobic bond, and they are responsible for docking score variation.

### 3.10. Molecular Dynamics

The molecular dynamics create a platform for verifying the docking procedure’s correctness in terms of the average root-mean-square deviation (RMSD), and root-mean-square fluctuation (RMSF), which reveal details regarding individual binding positions in the corresponding crystal structures [67]. We discovered that the root means square deviation (RMSD) of the docking refinement is less than 2.00 Å to become a better fitting pose of the ligand in the drug pocket and clearly demonstrates that the operating system is capable of effectively docking molecules [68,69]. RMSD measures the accuracy and reliability of a docking procedure by making the two docked poses parallel. Among the four proteins, the highest binding affinity was found for 2EDM (main protein); thus the molecular dynamic was performed against this protein.

The stability of these three docked complexes was evaluated using protein-ligand RMSD, ligand-protein interaction, hydrogen bonding, and ligand RMSF. In our study, the RMSD was calculated with respect to time (0–100 ns) and the interaction of amino acid residues of the protein. Firstly, it is noted that the RMSD is illustrated in Figure 6a to (f) in terms of time and amino acid residue where an innovative relationship is found for the first three Figure 6a–c. The RMSD was obtained within less than 2.0 Å, at 20 ns time, but it increased 2.5 Å at 50 ns time without any bond or interaction. However the RMSD did change after the formation of the backbone or hydrogen bond. The RMSD decreased from 2.5 Å to below 01 Å in terms of backbone bond interaction after docking, indicating a high accuracy and stability of docked complexes, but the hydrogen bonding shows aslight reduction in the RMSD value from the bond. It can be said that hydrogen bonds show little response to the molecular docking and stability of the docked complex. The RMSD shows at about 2.5 Å, but the protein-ligand interaction plays a significant role, showing the value at less than 1.0 Å when compound L02 had less than 0.7 Å. In the case of interaction with the amino acid residue and the ligand, the same phenomenon for the RMSD was obtained. 

The RMSF of the docked complex indicates stability. A lower value of RMSF mentions the higher strength. From Figure 6g, it has been found that the RMSF lays about 2.5 Å when it has no bonding or interaction as Ligand- protein interaction. In the case of hydrogen bond, it puts down 2.2 Å, which means that hydrogen bonds are little response for stability. But it has shifted down 01 Å due to backbone interaction, while the L02, L04, and L05 shows the minimum RMSF is about 0.7 Å, meaning the highest stability of the docked complex.

### 3.11. Trial Results

Different types of medication were applied, and their effectivenesses were checked. The dose depended on the size and weight of the fish. The name, quantity and performance of the drugs are listed in Table 9. In Table 9, the combination of OTC and vitamin C had the best performance when, at OTC 10.0 g per kg feed for seven days and vitamin C 10.0 g per kg feed for ten days, about 40.0–4.0% of the infected shrimp were cured, and shown in Figure 7. Afterwards, sulfadiazine (SFD) 10.0 g per kg of fish was administered for seven days, and it was noticed that about 5.0–7.0% of the fish were restored. Following that, several other medications were introduced progressively to broaden this study. The other medicines comprised oxytetracycline dehydrate (OTCD), and p-Mercapto-Sulfadiazine (p-M-SFD), and it was determined that oxytetracycline dehydrate had 10–15% effectiveness and p-Mercapto-Sulfadiazine (p-M-SFD) had 0.0% percent efficacy. The two medications were picked because they demonstrated promising outcomes separately, and, therefore, we wanted to test what would happen if they were used in combination. Accordingly, a combination of oxytetracycline 50% + sulfadiazine (OTC-5 g and SFD-5 g/kg feed for seven days) showed 30.0–35.0% effectiveness, and a combination of sulfadiazine with vitamin C showed less effectiveness at 3.0–5.0%. 

## 4. Conclusions

The goal of this research is to develop an effective medication or inhibitor against the deadly white spot disease caused by WSSV. Both computational and experimental tools performed as potential medication to treat WSSV. At the beginning of the study, some common inhibitors orally evaluated the molecular docking by calculating the binding affinity against four proteins. The highest molecular docking score was recorded for L01 (−7.0 kcal/mol) against the main protein 2FDM of WSSV, whereas the binding affinity of L01 in the evolved protein 2ED6 is at −6.4 kcal/mol, the highest was recorded as −6.6 kcal/mol for L04 against the evolved protein 2ED6. In the case of a drug’s existence in the market, L09, L10, L11 and L12 are showed the lower binding affinities than the newly designed drugs (L01 to L08) against both the evolved protein and main proteins. That is why, it is said that the eight derivatives of α-D-glucopyranoside (L01–L08) are more highly active for inhibition against WSSV than four established drugs available in the market (L09 to L12). Superior pharmacokinetic features, non-carcinogenicity, significant solubility in water, compliance with the Lipinski rule (except for L02, L03, L07, and L08), and drug-likeness features were also demonstrated for the potential drug candidates. Finally, molecular dynamics testing was carried out in order to validate its long-term persistence as a very promising medicinal candidate, and the results assessed the stability of the docked complexand acceptability as prospective new drug candidates. Furthermore, the rate of total drug clearance (renal and non-renal) was also excellent for all derivatives. Finally, a 40–45% cure rate was recorded by using the oxytetracycline (OTC) 50% with vitamin C for up to 10 days; the docking score as the binding affinity is lower than the modified derivatives of α-D-glucopyranoside. That is why it can be said that the WSSV is inhibited by modified derivatives of α-D-glucopyranoside than by oxytetracycline (OTC) or other existing drugs.

## Figures and Tables

**Figure 1 molecules-27-03694-f001:**
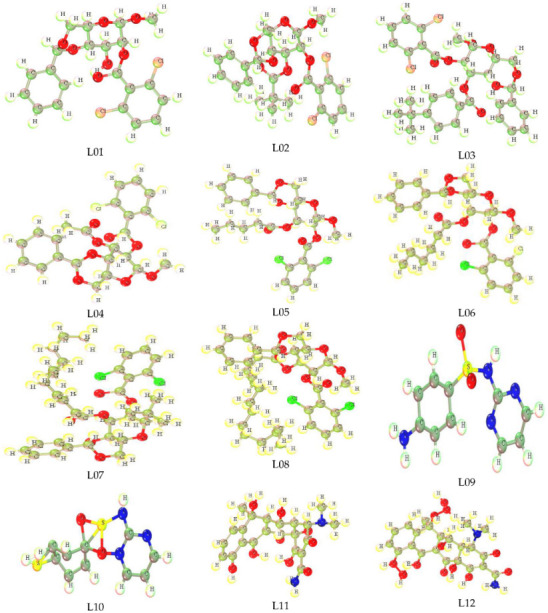
Optimized structure of inhibitors.

**Figure 2 molecules-27-03694-f002:**
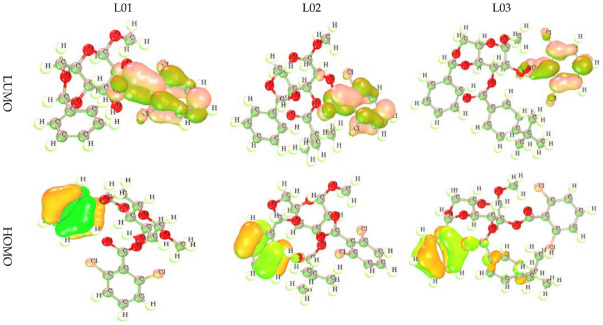
Frontier molecular orbitals diagram for HOMO and LUMO.

**Figure 3 molecules-27-03694-f003:**
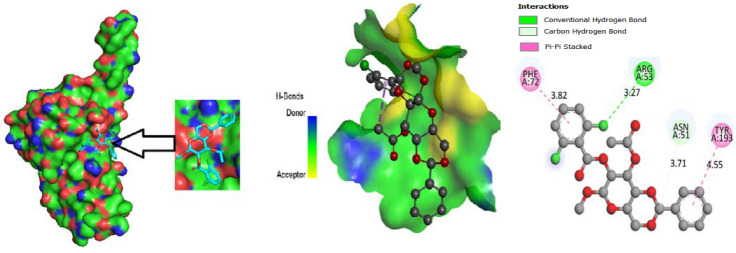
Molecular docking poses of envelope protein WSSV (PDB: 2ED6) with L04.

**Figure 4 molecules-27-03694-f004:**
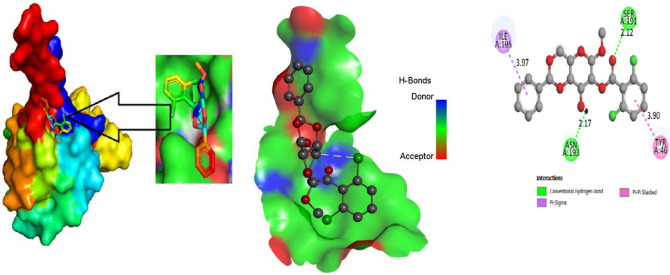
Molecular docking poses of white spot syndrome virus (PDB: 2GJI) with L01.

**Figure 5 molecules-27-03694-f005:**
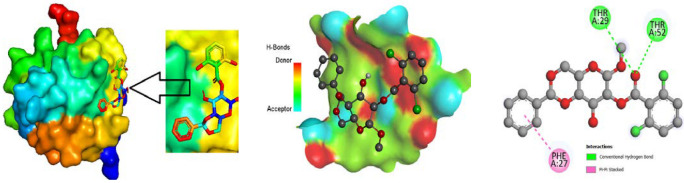
Molecular docking poses of white spot syndrome virus (PDB 2EDM) with L01.

**Figure 6 molecules-27-03694-f006:**
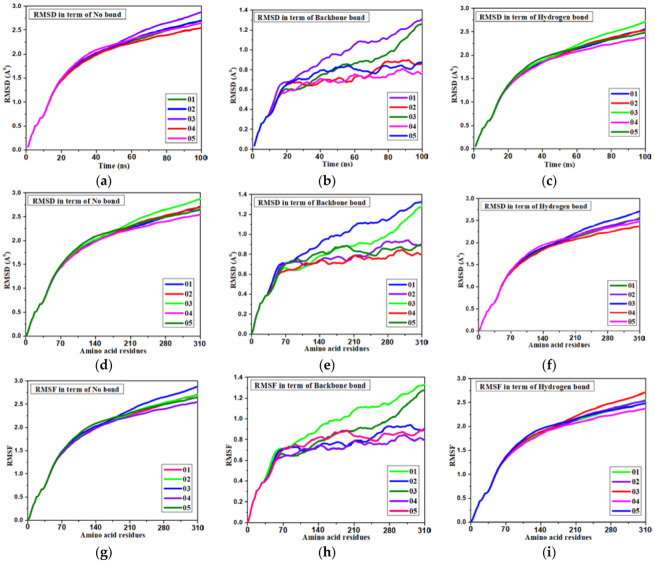
Various pictures of RMSD and RMSF for main protein (M^pro^) of white spot disease. (**a**) RMSD: time vs no bond. (**b**) RMSD: time vs protein skeleton. (**c**) RMSD: time vs hydrogen bond. (**d**) RMSD: amino acid vs no bond. (**e**) RMSD: amino acid vs backbone. (**f**) RMSD: amino acid vs H bond. (**g**) RMSF: amino acid vs no bond. (**h**) RMSF: amino acid vs backbone. (**i**) RMSF: amino acid vs H bond.

**Figure 7 molecules-27-03694-f007:**
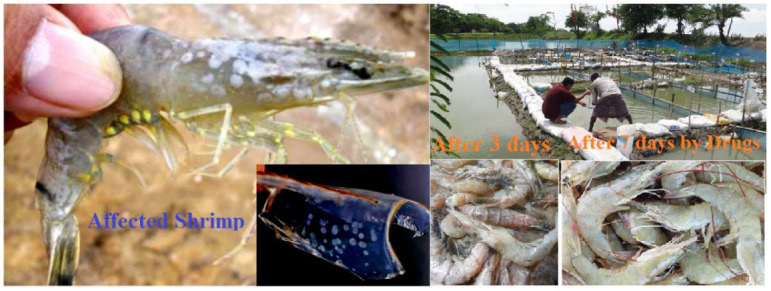
Trial results for the applied doses in WSD of shrimp.

**Table 1 molecules-27-03694-t001:** Protein information in shrimp white spot syndrome virus.

Title	PDB ID: 2ED6	PDB ID: 2GJ2	PDB ID: 2GJI	PDB ID: 2EDM
Organism	Shrimp white spot syndrome virus	Shrimp white spot syndrome virus	Shrimp white spot syndrome virus	Shrimp white spot syndrome virus
Resolution	2.00 Å	2.35 Å	N/A	2.20 Å
R-Value Free	0.281	0.275	N/A	0.278
References	[38]	[39]	[39]	[38]

**Table 2 molecules-27-03694-t002:** Grid box parameters used for docking analysis in this study for white spot disease (WSD).

Protein Name with PDB ID	Grid Box Size	
	Center	Dimension (Å)
Envelope Protein WSSV (PDB 2ED6)	X = 28.2583	X = 38.9011
	Y = 106.048	Y = 67.0482
	Z = 92.9776	Z = 45.525
White Spot Syndrome Virus (PDB 2GJI)	X = −8.6514	X = 32.888
	Y = 15.6227	Y = 33.828
	Z = −5.5754	Z = 43.396
White Spot Syndrome Virus (PDB 2EDM)	X= 37.1819	X= 39.3455
	Y= 35.3181	Y= 44.655
	Z= 92.9466	Z= 61.178
White Spot Syndrome Virus (PDB 2GJ2)	X = 36.2550	X = 34.8480
	Y = 1.4367	Y = 37.8846
	Z = −6.1508	Z= 28.8952

**Table 3 molecules-27-03694-t003:** Data of chemical descriptors.

Ligand.	LUMO	HOMO	A = −LUMO	I = −HOMO	Energy Gap = I − A	$\mathrm{Chemical}\;\mathrm{Potential}\;\left(\mu \right)=-\frac{I+A}{2}$	Hardness $\left(\eta \right)=\frac{I-A}{2}$	$\mathrm{Electronegativity}\;\;\;\;\;\left(\chi \right)=\frac{I+A}{2}$	$\mathrm{Softness}\left(\sigma \right)=\frac{1}{\eta }$	Electrophilicity $\left(\omega \right)=\frac{\mu^{2}}{2\eta }$
L01	−1.685	−8.445	1.685	8.445	6.760	−5.065	3.38	5.065	0.2959	3.7950
L02	−1.550	−8.528	1.550	8.528	6.978	−5.039	3.489	5.039	0.2866	3.6388
L03	−1.413	−8.159	1.413	8.159	6.746	−4.786	3.373	4.786	0.3965	3.3955
L04	−1.647	−8.896	1.647	8.896	7.249	−5.2715	3.6245	5.271	0.2759	3.8335
L05	−1.594	−8.837	1.594	8.837	7.243	−5.2155	3.6215	5.215	0.2761	3.7555
L06	−1.701	−8.605	1.701	8.605	6.904	−5.1530	3.452	5.153	0.2897	3.8461
L07	−1.580	−8.573	1.580	8.573	6.993	−5.0765	3.4965	5.076	0.2860	3.6852
L08	−1.624	−8.909	1.624	8.909	7.285	−5.2665	3.6425	5.266	0.2745	3.8073
L09	−0.68	−8.673	0.68	8.673	7.993	−4.6765	3.9965	4.6765	0.2503	2.7361
L10	−1.240	−7.877	1.240	7.877	6.637	−4.5585	3.3185	4.5585	0.3013	3.1309
L11	−2.163	−9.146	2.163	9.146	6.983	−5.6545	3.4915	5.6545	0.2864	4.5787
L12	−1.745	−8.464	1.745	8.464	6.719	−5.1045	3.3595	5.1045	0.2977	3.8779

**Table 4 molecules-27-03694-t004:** Data of binding energy and name of interacted ligand against WSSV in binding affinity (kcal/mol).

Ligands	Envelope Protein WSSV (PDB ID: 2ED6)	Main Protease of WSSV (PDB ID: 2GJ2)	Main Protease of WSSV (PDB ID: 2GJI)	Main Protease of WSSV (PDB ID: 2EDM)
L01	−6.4	−6.20	−6.2	−7.0
L02	−5.6	−6.30	−6.0	−6.4
L03	−6.5	−6.90	−6.1	−6.6
L04	−6.6	−6.20	−5.7	−6.3
L05	−5.6	−6.20	−5.7	−6.0
L06	−6.2	−5.80	−5.5	−6.1
L07	−5.6	−5.80	−5.0	−5.7
L08	−5.5	−5.10	−4.7	−5.5
L09	−5.6	−6.54	−5.9	−5.7
L10	−5.7	−6.74	−5.6	−5.4
L11	−6.1	−6.4	−6.4	−6.4
L12	−5.5	−5.4	−5.1	−5.8

**Table 5 molecules-27-03694-t005:** Data of inhibition constant, binding energy, efficiency, and total energy of WSSV.

Ligand	Inhibitor Constant (µM)	Ligand Efficiency (kcal/mol)	Internal Energy (kcal/mol)	Electrostatic Energy (kcal/mol)	Total Internal Energy (kcal/mol)	Torsional Energy (kcal/mol)	Unbound Energy (kcal/mol)
L01	40.00	−0.19	−7.37	−0.15	−2.60	1.79	−2.60
L02	36.00	−0.17	−8.44	−0.24	−2.73	2.39	−2.73
L03	10.28	−0.15	−9.06	−0.09	−4.49	2.68	−4.49
L04	16.21	−0.38	−7.43	−1.40	−1.58	0.89	−1.58
L05	11.43	−0.40	−7.64	−1.37	−0.27	0.89	−0.27
L06	17.18	−0.43	−6.88	−0.46	−3.71	2.45	−2.08
L07	17.25	−0.36	−6.69	−0.64	−4.10	2.32	−1.94
L08	18.66	−0.32	−6.51	−0.69	−3.98	2.11	−1.96
L09	22.23	−0.22	−6.34	−0.71	−3.78	2.67	−1.76
L10	21.23	−0.24	−6.10	−0.88	−3.67	3.20	−2.44
L11	20.01	−0.26	−5.90	−0.81	−4.20	2.45	−2.87
L12	18.56	−0.28	−6.10	−0.96	−4.36	2.98	2.62

**Table 6 molecules-27-03694-t006:** Data of Lipinski rule, pharmacokinetics, and drug likeness.

Ligands	NBR	HBA	HBD	TPSA, Å²	Consensus Log Po/w	Log Kp (Skin Permeation), cm/s	Lipinski Rule		MW	Bioavailability Score	GI Absorption
							Result	Violation			
L01	05	07	01	83.45	3.16	−6.69	Yes	00	455.29	0.55	High
L02	08	08	0	89.52	4.23	−5.81	No	01	539.40	0.55	High
L03	09	08	0	89.52	5.64	−4.90	No	02	615.50	0.17	Low
L04	07	08	00	89.52	3.37	−6.54	Yes	00	497.32	0.55	High
L05	10	08	00	89.52	4.47	−5.83	Yes	01	539.40	0.55	High
L06	11	08	00	89.52	4.75	−5.53	Yes	01	553.43	0.55	High
L07	06	07	00	80.29	7.79	−2.93	No	02	649.65	0.17	Low
L08	19	08	00	89.52	7.56	−3.13	No	02	665.64	0.17	Low
L09	03	06	02	109.47	−0.26	−8.16	Yes	0	260.36	0.55	High
L10	03	04	01	119.13	1.18	−7.41	Yes	0	267.33	0.55	High
L11	02	10	07	201.85	−1.04	−9.62	No	02	460.43	0.11	Low
L12	N/A	N/A	N/A	N/A	N/A	N/A	N/A	N/A	528.16	N/A	N/A

**Table 7 molecules-27-03694-t007:** Data of the ADME properties.

Ligands.	Caco-2 Permeability	Blood Brain Barrier Permeant	P-I Glycoprotein Inhibitor	P-Glycoprotein Substrate	Total Clearance	CYP2C9 Inhibitor	CYP 1A2 Inhibitor
L01	1.47	No	Yes	No	0.595	No	No
L02	1.807	No	Yes	No	0.431	No	No
L03	1.878	No	Yes	No	0.424	Yes	No
L04	1.70	No	Yes	No	0.561	No	No
L05	0.747	No	No	No	0.711	No	Yes
L06	1.778	No	Yes	No	0.627	Yes	Yes
L07	1.758	No	Yes	No	0.705	No	No
L08	1.59	No	Yes	No	0.873	No	No
L09	−0.018	No	No	No	0.642	No	No
L10	1.296	No	No	No	−0.112	No	No
L11	−0.538	No	No	Yes	0.456	No	No
L12	−0.595	N/A	No	Yes	0.225	No	No

**Table 8 molecules-27-03694-t008:** Aquatic and non-aquatic toxicity.

Ligands	Max Tolerated Dose (mg/kg/day)	Oral Rat Chronic Toxicity ((LOAEL)	Hepatotoxicity	AMES Toxicity	Water Solubility, Log S	Oral Rat Acute Toxicity (LD50) (mol/kg)	T. Pyriformis Toxicity (log μg/L)
L01	0.581	1.556	No	No	−4.658	2.746	0.285
L02	0.674	1.530	No	No	−4.219	3.034	0.285
L03	0.590	1.113	No	No	−3.698	2.910	0.285
L04	0.822	1.522	No	No	−4.674	3.264	0.285
L05	0.438	10.30	No	Yes	−2.892	2.482	0.285
L06	0.763	1.524	No	No	−5.509	3.148	0.285
L07	0.525	1.396	No	No	−4.321	2.302	0.285
L08	0.700	1.497	No	No	−5.021	2.621	0.285
L09	1.156	1.97	Yes	No	−2.954	2.234	0.285
L10	1.014	1.838	Yes	No	−3.076	2.348	0.285
L11	1.136	5.156	No	No	−2.528	5.156	0.285
L12	1.045	4.524	No	No	−2.497	2.456	0.285

**Table 9 molecules-27-03694-t009:** Dose and performance of the drugs.

S.L. No.	Name of the Drugs	Dose	Cure Rate
1	Oxytetracycline (OTC) 50% with vitamin C	OTC-10 g/kg feed for 7 days and VC 10 g/kg feed for 10 days	40–45%
2	Sulfadiazine (SFD)	10 g/kg feed for 7 days	5–7%
3	Oxytetracycline dehydrate (OTCD)	10 g/kg feed for 7 days	10–15%
4	p-Mercapto-Sulfadiazine (p-M-SFD)	10 g/kg feed for 7 days	00%
5	Oxytetracycline 50%+ sulfadiazine	OTC-5 g and SFD-5 g/kg feed for 7 days	30–35%
6	Sulfadiazine with vitamin	SFD-10 g/kg feed and VC 10 g/kg feed for 7 days	3–5%

## Data Availability

Not applicable.

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
