# Peer review of "Investigation of the New Inhibitors by Sulfadiazine and Modified Derivatives of α-D-glucopyranoside for White Spot Syndrome Virus Disease of Shrimp by In Silico: Quantum Calculations, Molecular Docking, ADMET and Molecular Dynamics Study"

_molecules, 2022, doi:10.3390/molecules27123694_

Round 1

Reviewer 1 Report

This is Investigation of the new inhibitors by Sulfadiazine and modified derivatives of α-D-glucopyranoside for White Spot Syndrome Virus disease of Shrimp by In sillico. The results are interesting the work has been performed carefully. I have just some suggestions for the authors before the acceptance the manuscript

General comments

  • In section 2.1. Optimization and ligand preparation

It is important to mention that the reactivity parameters were determined by orbital approximation

Line 138

Have a type error in letter “M”, please change for “µ”

  • In section 2.2. Protein preparation and collection

Line 143 “It´s necessary to put the page and consultation date of PDB (http://www.rcsb.org, accessed on XX may 2022).

  • In section 2.3. Molecular Docking and visualization of docking

Lines 152-154

  • It´s necessary to mention what the PyRx software consists. Also, its recommended to including which approach used "AutoDock 4 or Autodock vina".
  • Please provide the detail Autodock calculations.
  • It´s important mentioned “the partial charges were added and non-polar hydrogen atoms” by the PyRx software or for the Autodock software used.

Lines 157-158

In table 2 in the dimensions column please attached the value of the z dimension for (PDB 2GJ2) protein.

  • In section 2.5. Lipinski Rule and Pharmacokinetics

Line 170

Have a type error in letter “Å2”, please change the number “2” for superscript (Å2)

  • In section 2.6. Molecular Dynamic

Line 174

It is necessary to put what MD means and not only their acronyms

  • In section 3.1. Optimized structure

Lines 197-201

It is necessary to improve the paragraph

It is important for authors explain how determinate the structural shape optimization using Bézier triangles in the geometry optimization in the inhibitors according to reference used [51].

Line 202

Change the phase "figure 01" for "figure 1"

Line 203

It is necessary to improve the Figures “Methyl 4,6-O-benzylidene-2-O-(2,6-dichloroben-zoyl)-3-O-pivaloyl-α-D-glucopyranoside (02), Methyl 4,6-O-benzylidene-2-O-(2,6-dichlorobenzoyl)-3-O-myristoyl-α-D-glucopyranoside (08), Oxytetracycline (11)”, because some atoms are not observed.

  • In section 3.2. HOMO, LUMO, and chemical reactivity descriptors

Line 207

Change the phase "table 03" for "table 3"

Please change this phrase “In general, the higher the hardness value of a material means that the object has greater strength” to “In general, the higher the hardness value of these compounds means that strongly the molecules resist changes electron configuration”

Check these articles

doi.org/10.1007/s00894-018-3868-4

doi.org/10.3390/computation7030052

doi.org/10.1021/j100023a006

Line 221

Recommend attached L01, L02…L12 in the S/N column in the table 3, so that the information in the table matches the description of lines 212 and 215.

Change "S/N" for "Ligand"

  • In section 3.3. Frontier Molecular orbital: HOMO and LUMO

Line 229

Change the phase "figure 02" for "figure 2"

Line 249

Change the phase "table 04" for "table 4"

Lines 244-245

The same recommendation attached L01, L02…L12 in the Ligand column in the table 4, so that the information in the table matches the description of line 250.

Lines 244-245

Change this phrase “Sulfadiazine-d4, p-Mercapto-sulfadiazine, Oxytetracycline, and Oxytetracycline Hydrate” to “L09, L10, L11 and L12” for homologue.

Line 259

The same recommendation attached L01, L02…L12 in the Ligand column in the table 5

  • In the section 3.5. Protein-Ligands Interaction

Line 268

Delate a one dot

  • In the section 3.6. Pharmacokinetics and Drug likeness study

Lines 284-285

Please change the identification of the ligands “Ligand 03, 07 and 08” for “Ligand L03,L07 and L08”, the seme case for the line 285.

Line 287

Change the phase "table 06" for "table 6"

Lines 244-245

The same recommendation attached L01, L02…L12 in the Ligand column in the table 6, and also attached the name ligand this column.

  • In the section 3.7. Pharmacokinetics and ADMET studies

Lines 290 and 293

Change the phrase “In silico” for “In silico”

Line 295

Change the phase "table 07" for "table 7"

Recommend review the identification of table 7 and table 8. Improve the quality these tables

  • In the section 3.8. Aquatic and non-aquatic Toxicity

Line 308

Change the phase "table 08" for "table 8"

  • In the section Change the phase "table 07" for "table 7"

Line 333

Because ligands 9-12 were not reported in table 9

  • In section 3.10. Protein-ligand interaction

Line 336

Change the phase "figure 03" for "figure 3"

The Figure 4 and 5 are not mentioned in this section

  • In section 3.11. Molecular Dynamics

The Figure 6 is not mentioned in this section

  • In section 3.12. Trial results

The Figure 7 is not mentioned in this section

Constructive criticism

Please homologate the identification of the ligands “L01, L02, L03 ……. L12” in all tables and article

Author Response

General comments

Reviewer 1:

Q1: In section 2.1. Optimization and ligand preparation

It is important to mention that the reactivity parameters were determined by orbital approximation

Line 138, Have a type error in letter “M”, please change for “µ”

Answer: It has done.

Q2: In section 2.2. Protein preparation and collection.

Line 143 “It´s necessary to put the page and consultation date of PDB (http://www.rcsb.org, accessed on XX may 2022).

Answer: It has done.

Q3: In section 2.3. Molecular Docking and visualization of docking

Lines 152-154, It´s necessary to mention what the PyRx software consists. Also, its recommended to including which approach used "AutoDock 4 or Autodock vina".

Answer: It has done.

Q4: Please provide the detail Autodock calculations.

Answer: It has added the all details in table 2 for this calculation.

Q5: It´s important mentioned “the partial charges were added and non-polar hydrogen atoms” by the PyRx software or for the Autodock software used.

Lines 157-158, in table 2 in the dimensions column please attached the value of the z dimension for (PDB 2GJ2) protein.

Answer: It has added the all details in table 2.

Q6: In section 2.5. Lipinski Rule and Pharmacokinetics

Line 170, Have a type error in letter “Å2”, please change the number “2” for superscript (Å2).

Answer: It has done.

Q7: In section 2.6. Molecular Dynamic

Line 174, It is necessary to put what MD means and not only their acronyms.

Answer: It has done.

Q8: In section 3.1. Optimized structure

Lines 197-201,  It is necessary to improve the paragraph

Answer: It has done.

Q9: It is important for authors explain how determinate the structural shape optimization using Bézier triangles in the geometry optimization in the inhibitors according to reference used [51].

Answer: It has done.

Q10: Line 202, Change the phase "figure 01" for "figure 1"

Answer: It has done.

Q11: Line 203, It is necessary to improve the Figures “Methyl 4,6-O-benzylidene-2-O-(2,6-dichloroben-zoyl)-3-O-pivaloyl-α-D-glucopyranoside (02), Methyl 4,6-O-benzylidene-2-O-(2,6-dichlorobenzoyl)-3-O-myristoyl-α-D-glucopyranoside (08), Oxytetracycline (11)”, because some atoms are not observed.

Answer: It has done. If two molecules are added in one row, it will be more visible but need huge space.

Q12: In section 3.2. HOMO, LUMO, and chemical reactivity descriptors

Line 207, Change the phase "table 03" for "table 3"

Answer: It has done.

Q13: Please change this phrase “In general, the higher the hardness value of a material means that the object has greater strength” to “In general, the higher the hardness value of these compounds means that strongly the molecules resist changes electron configuration”

Answer: It has done.

Q14: Check these articles

doi.org/10.1007/s00894-018-3868-4

doi.org/10.3390/computation7030052

doi.org/10.1021/j100023a006

Answer: It has done and added these as references.

Q15: Line 221, Recommend attached L01, L02…L12 in the S/N column in the table 3, so that the information in the table matches the description of lines 212 and 215.

Change "S/N" for "Ligand"

Answer: It has done and added.

Q15: In section 3.3. Frontier Molecular orbital: HOMO and LUMO

Line 229, Change the phase "figure 02" for "figure 2"

Answer: It has done and added.

Q16: Line 249, Change the phase "table 04" for "table 4"

Answer: It has done and added

Q17: Lines 244-245, The same recommendation attached L01, L02…L12 in the Ligand column in the table 4, so that the information in the table matches the description of line 250.

Answer: It has done and added.

Q18: Lines 244-245, Change this phrase “Sulfadiazine-d4, p-Mercapto-sulfadiazine, Oxytetracycline, and Oxytetracycline Hydrate” to “L09, L10, L11 and L12” for homologue.

Answer: It has done and added.

Q19: Line 259, The same recommendation attached L01, L02…L12 in the Ligand column in the table.

Answer: It has done and added.

Q20: In the section 3.5. Protein-Ligands Interaction

Line 268, Delate a one dot

Answer: It has done and added.

Q21: In the section 3.6. Pharmacokinetics and Drug likeness study

Q22: Lines 284-285, Please change the identification of the ligands “Ligand 03, 07 and 08” for “Ligand L03,L07 and L08”, the seme case for the line 285.

Answer: It has done and added.

Q23: Line 287, Change the phase "table 06" for "table 6".

Answer: It has done and added.

Q24: Lines 244-245, The same recommendation attached L01, L02…L12 in the Ligand column in the table 6, and also attached the name ligand this column.

Answer: It has done and added.

Q25: In the section 3.7. Pharmacokinetics and ADMET studies

Lines 290 and 293, Change the phrase “In silico” for “In silico”

Answer: It has done and added.

Line 295, Change the phase "table 07" for "table 7"

Q26: Recommend review the identification of table 7 and table 8. Improve the quality these tables.

Answer: It has done and added.

Q27: In the section 3.8. Aquatic and non-aquatic Toxicity

Line 308, Change the phase "table 08" for "table 8"

Answer: It has done and added.

Q28: In the section Change the phase "table 07" for "table 7"

Line 333, Because ligands 9-12 were not reported in table 9

Answer: It has done and added.

Q29:  In section 3.10. Protein-ligand interaction

Line 336, Change the phase "figure 03" for "figure 3"

Answer: It has done and added.

The Figure 4 and 5 are not mentioned in this section

Q30: In section 3.11. Molecular Dynamics

The Figure 6 is not mentioned in this section

Answer: It has done and added.

Q31: In section 3.12. Trial results

The Figure 7 is not mentioned in this section

Answer: It has done and added.

Q31: Constructive criticism

Please homologate the identification of the ligands “L01, L02, L03 ……. L12” in all tables and article

Answer: Yes, it made the all ligand identification of the ligands “L01, L02, L03 ……. L12 in full manuscript.

Reviewer 2 Report

Comments:

The design of this paper is relatively reasonable and reliable. Sulfadiazine and modified derivatives of É‘-D-glucopyranoside ’s inhibition for White Spot Syndrome Virus (WSSV) disease of Shrimp were investigated by in sillico. The experimental logic is rigorous, and the data obtained is convincing. In this paper, the proper protein were chose by optimization and ligand preparation. The activity of sulfadiazine and modified derivatives of É‘-D-glucopyranoside for White Spot Syndrome Virus disease of Shrimp were assessed by Molecular Docking, Pharmacokinetics, ADMET,  Molecular Dynamic and quantitative structure–activity relationship (QSAR) studies.   Finally, the selected drugs were tested by White Spot trial procedure for their effect.  This research has certain theoretical significance and practical value. However, this thesis still has some problems.

  1. In Section 3.2 and Table3, the result of HOMO -LUMO gap for ligand 09(L09) 7.993 is maximum, but there is a sentence ‘the HOMO–LUMO gap ranges between 6.760 and 8.808 eV for all studied chemicals, with L04, L08 and L09, having the minor energy gap and the minimum softness value ’. There is ambiguity, please give an explanation.

  1. In Table 10, only 6 drugs were chosen to test activity for White Spot Syndrome Virus disease of Shrimp, but the reason about why choose these drugs from 12 compounds which were studied by in sillico was not state in any part. Please state it clearly.

  1. In section 4, author present a conclusion that the Oxytetracycline(OTC) 50% with vitamin C which docking score as binding affinity is lower than modified derivatives of É‘-D-glucopyranoside has effective cure rate so that it can be said that the WSSV is more inhibited by modified derivatives of É‘-D-glucopyranoside than Oxytetracycline (OTC) or other existing drugs. The evidence of this conclusion is not enough, please give more evidences in this article.

Author Response

Comments:

The design of this paper is relatively reasonable and reliable. Sulfadiazine and modified derivatives of É‘-D-glucopyranoside ’s inhibition for White Spot Syndrome Virus (WSSV) disease of Shrimp were investigated by in sillico. The experimental logic is rigorous, and the data obtained is convincing. In this paper, the proper protein were chose by optimization and ligand preparation. The activity of sulfadiazine and modified derivatives of É‘-D-glucopyranoside for White Spot Syndrome Virus disease of Shrimp were assessed by Molecular Docking, Pharmacokinetics, ADMET,  Molecular Dynamic and quantitative structure–activity relationship (QSAR) studies.   Finally, the selected drugs were tested by White Spot trial procedure for their effect.  This research has certain theoretical significance and practical value. However, this thesis still has some problems.

Q1: In Section 3.2 and Table3, the result of HOMO -LUMO gap for ligand 09(L09) 7.993 is maximum, but there is a sentence ‘the HOMO–LUMO gap ranges between 6.760 and 8.808 eV for all studied chemicals, with L04, L08 and L09, having the minor energy gap and the minimum softness value ’. There is ambiguity, please give an explanation.

   Answer: It is revised and added with rewrite.

 Q2. In Table 09, only 6 drugs were chosen to test activity for White Spot Syndrome Virus disease of Shrimp, but the reason about why choose these drugs from 12 compounds which were studied by in sillico was not state in any part. Please state it clearly.

   Answer: It is revised and added in table 09..

Q3. In section 4, author present a conclusion that the Oxytetracycline(OTC) 50% with vitamin C which docking score as binding affinity is lower than modified derivatives of É‘-D-glucopyranoside has effective cure rate so that it can be said that the WSSV is more inhibited by modified derivatives of É‘-D-glucopyranoside than Oxytetracycline (OTC) or other existing drugs. The evidence of this conclusion is not enough; please give more evidences in this article.

Answer: For the more evidences, the molecular docking, pass prediction, ADME, toxicity and Molecular dynamic were performed for all compounds. As it is the in silico study, the existence drugs in market were performed the trial of experimental study which is further compared with the in silico study among all compounds which is the main objective of this study. Finally, the conclusion was rewrote and revised.

Reviewer 3 Report

The paper described " Investigation of the new inhibitors by Sulfadiazine and modified derivatives of α-D-glucopyranoside for White Spot Syndrome Virus disease of Shrimp by in sillico: Quantum calculations, molecular docking, ADMET, and molecular dynamics study" in great details and design. This should be a good reference for other viewers who are interested in such subject. Therefore, I am in favor of accepting this paper for publishing. However, there are still some minor comments that should be addressed:

1-    1- The work design or rationale is not clear. Why the authors select D-glucopyranoside and its derivatives for the treatment of WSSV?

2-    2- Why the authors did not try experimental testing for D-glucopyranoside and its derivatives activity?

3-    3- The authors mentioned in line 62 that the 5th position has been placed for Bangladesh in the world of fish production, however, the authors cited a very old reference (13, 2001) Please update this data.

4-    4- In line 102: what is meant by “obtaining the shore and white spots”?

5-    5- In table 1, no need to repeat the organism type in the 4 columns. You can just mention it in the table title, e.g., Table 1. Protein Information in Shrimp white spot syndrome virus.

6-    6- Provide figures 3-5 in a high resolution.

7-    7- Please check well the whole manuscript, there are many grammar issues, for example:

a)     In line 85, change “detected on 2001” to “detected in 2001”. The preposition "on" is used for specific dates and for days of the week.

b)    In line 86, is it warming or warning?

c)     Check the grammar and rephrase the following sentence: Among them, the White spot disease, detected on 2001 in Bangladesh, has been considered as warming for shrimp farming of Bangladesh due to severe functions to destroy the full farms due to forwarding in death to the host body.

d)    Please correct 50 in PIC50 to be subscript, line 188.

e)     In section 3.11. Molecular Dynamics, correct the sentence: “the slight reduction of RMDS value from on bond”.

f)     In conclusion section: “white spot disease” NOT “white sport disease.”

g)    In conclusion section: At first time, we have studied by both computational and experimental tools

h)    In conclusion section: correct “were taken for in sillioc study”

Author Response

The paper described " Investigation of the new inhibitors by Sulfadiazine and modified derivatives of α-D-glucopyranoside for White Spot Syndrome Virus disease of Shrimp by in sillico: Quantum calculations, molecular docking, ADMET, and molecular dynamics study" in great details and design. This should be a good reference for other viewers who are interested in such subject. Therefore, I am in favor of accepting this paper for publishing. However, there are still some minor comments that should be addressed:

Q1- The work design or rationale is not clear. Why the authors select D-glucopyranoside and its derivatives for the treatment of WSSV?

Answer: The main motivation and objectives of this study rewrote and added in the introduction section.

Q2- Why the authors did not try experimental testing for D-glucopyranoside and its derivatives activity?

 Answer: Actually, it is the in silico study for the D-glucopyranoside and its derivatives activity against treatment of WSSV. That is why, it is no option to perform the experimental work. Moreover, we conducted the trial of market existence drugs and compare their result with in silico.

Q3- The authors mentioned in line 62 that the 5th position has been placed for Bangladesh in the world of fish production, however, the authors cited a very old reference (13, 2001) Please update this data.

Answer: It has done and added. After that there is no more study for citing. So, we cannot add the more study next updated citations. However, the new citation “HABIB, T. B., Bangladesh sees significant rise in fish production in a decade (https://thefinancialexpress.com.bd/trade/bangladesh-sees-significant-rise-in-fish-production-in-a-decade-1636256725). The Financial Express November 07, 2021” were added in number 14.

Q4- In line 102: what is meant by “obtaining the shore and white spots”?

Answer: It was corrected, there are typing errors.

Q5- In table 1, no need to repeat the organism type in the 4 columns. You can just mention it in the table title, e.g., Table 1. Protein Information in Shrimp white spot syndrome virus.

Answer: It was corrected and added.

Q6- Provide figures 3-5 in a high resolution.

Answer: It was corrected and added.

Q7- Please check well the whole manuscript, there are many grammar issues, for example:

  1. In line 85, change “detected on 2001” to “detected in 2001”. The preposition "on" is used for specific dates and for days of the week.

Answer: It was corrected and added.

  1. In line 86, is it warming or warning?

Answer: It was corrected and added.

  1. c)     Check the grammar and rephrase the following sentence: Among them, the White spot disease, detected on 2001 in Bangladesh, has been considered as warming for shrimp farming of Bangladesh due to severe functions to destroy the full farms due to forwarding in death to the host body.
  2. d)    Please correct 50 in PIC50 to be subscript, line 188.

Answer: It was corrected and added.

  1. e)     In section 3.11. Molecular Dynamics, correct the sentence: “the slight reduction of RMDS value from on bond”.

Answer: It was corrected and added.

  1. f)     In conclusion section: “white spot disease” NOT “white sport disease.”

Answer: It was corrected and added.

  1. g)    In conclusion section: At first time, we have studied by both computational and experimental tools.

Answer: It was corrected and added.

  1. h)    In conclusion section: correct “were taken for in sillioc study”.

Answer: It was corrected and added.
